# Evidence for intrinsic charm quarks in the proton

The NNPDF Collaboration*

The theory of the strong force, quantum chromodynamics, describes the proton in terms of quarks and gluons. The proton is a state of two up quarks and one down quark bound by gluons, but quantum theory predicts that in addition there is an infinite number of quark–antiquark pairs. Both light and heavy quarks, whose mass is respectively smaller or bigger than the mass of the proton, are revealed inside the proton in high-energy collisions. However, it is unclear whether heavy quarks also exist as a part of the proton wavefunction, which is determined by non-perturbative dynamics and accordingly unknown: so-called intrinsic heavy quarks[1]. It has been argued for a long time that the proton could have a sizable intrinsic component of the lightest heavy quark, the charm quark. Innumerable efforts to establish intrinsic charm in the proton[2] have remained inconclusive. Here we provide evidence for intrinsic charm by exploiting a high-precision determination of the quark–gluon content of the nucleon[3] based on machine learning and a large experimental dataset. We disentangle the intrinsic charm component from charm–anticharm pairs arising from high-energy radiation[4]. We establish the existence of intrinsic charm at the 3-standard-deviation level, with a momentum distribution in remarkable agreement with model predictions[1,5]. We confirm these findings by comparing them to very recent data on $Z$-boson production with charm jets from the Large Hadron Collider beauty (LHCb) experiment[6].

The foundational deep-inelastic scattering experiments at the SLAC linear collider in the late 1960s and early 1970s demonstrated the presence inside the proton of point-like constituents, soon identified with quarks, the elementary particles that interact and are bound inside the proton by gluons, the carriers of the strong nuclear force. It was rapidly clear, and confirmed in detail by subsequent studies, that these point-like constituents, collectively called partons by Feynman[7], include the up and down quarks that carry the proton quantum numbers, but also gluons, as well as an infinite number of pairs of quarks and their antimatter counterparts, antiquarks. The description of electron–proton and proton–proton collisions at high momentum transfers in terms of collisions between partons is now rooted in the theory of quantum chromodynamics (QCD), and it provides the basis of modern-day precision phenomenology at proton accelerators such as the Large Hadron Collider (LHC) of CERN[8] as well as for future facilities including the Electron–Ion Collider[9], the Forward Physics Facility[10] and neutrino telescopes[11].

Knowledge of the structure of the proton, which is necessary to obtain quantitative prediction for physics processes at the LHC and other experiments, is encoded in the distribution of momentum carried by partons of each type (gluons, up quarks, down quarks, up antiquarks and so on): parton distribution functions (PDFs). These PDFs could, in principle, be computed from first principles, but in practice even their determination from numerical simulations[12] is extremely challenging. Consequently, the only strategy available at present for obtaining the reliable determination of the proton PDFs that is required to evaluate LHC predictions is empirical, through the global analysis of data for which precise theoretical predictions and experimental measurements are available, so that the PDFs are the only unknown[8].

Although this successful framework has by now been worked through in great detail, several key open questions remain open. One of the most controversial of these concerns the treatment of so-called heavy quarks (that is, those whose mass is greater than that of the proton; $m_p = 0.94$ GeV). Indeed, virtual quantum effects and energy–mass considerations suggest that the three light quarks and antiquarks (up, down and strange) should all be present in the proton wavefunction. Their PDFs are therefore surely determined by the low-energy dynamics that controls the nature of the proton as a bound state. However, it is well known[8,13–15] that in high enough energy collisions all species of quarks can be excited and hence observed inside the proton, so their PDFs are nonzero. This excitation follows from standard QCD radiation, and it can be computed accurately in perturbation theory.

However, then the question arises of whether heavy quarks also contribute to the proton wavefunction. Such a contribution is called intrinsic, to distinguish it from that computable in perturbation theory, which originates from QCD radiation. Already since the dawn of QCD, it was argued that all kinds of intrinsic heavy quark must be present in the proton wavefunction[16]. In particular, it was suggested[1] that the intrinsic component could be non-negligible for the charm quark, whose mass ($m_c \simeq 1.51$ GeV) is of the same order of magnitude as the mass of the proton.

This question has remained highly controversial, and indeed recent dedicated studies have resulted in disparate claims, from excluding

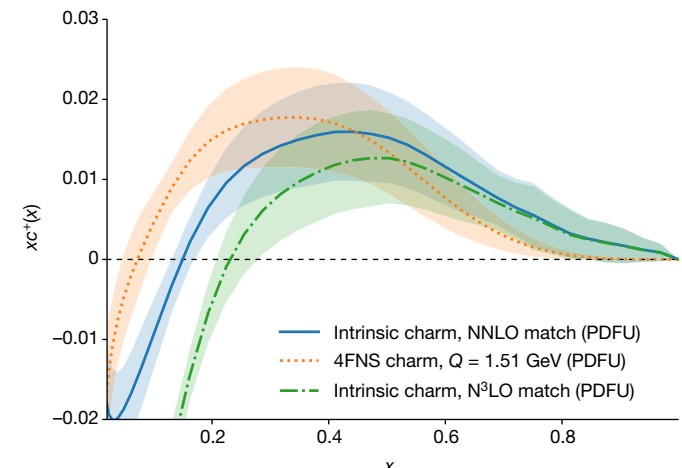

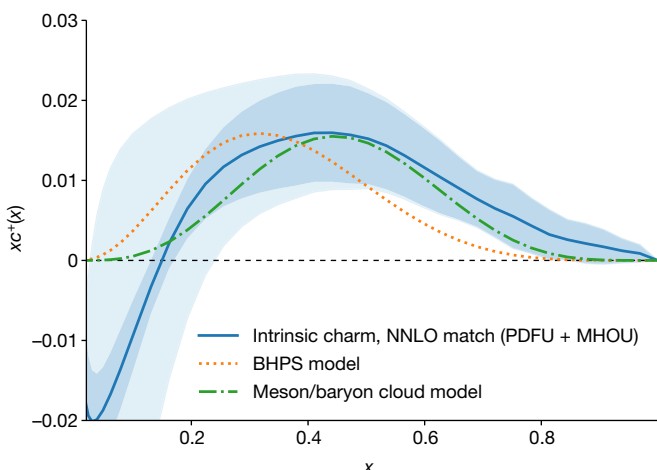

**Fig. 1 | The intrinsic charm PDF and comparison with models.** Left, the purely intrinsic (3FNS) result (blue) with PDFU alone, compared to the 4FNS PDF, which includes both an intrinsic and a radiative component, at $Q = m_c = 1.51\,\text{GeV}$ (orange). The purely intrinsic (3FNS) result obtained using N³LO matching is also shown (green). Right, the purely intrinsic (3FNS) final result with total uncertainty (PDFU + MHOU), with the PDFU indicated as a dark shaded band; the predictions from the original BHPS model[1] and from the more recent meson/baryon cloud model[5] are also shown for comparison (dotted and dot-dashed curves, respectively).

momentum fractions carried by intrinsic charm larger than 0.5% at the 4-standard-deviation (4σ) level[17] to allowing up to a 2% charm momentum fraction[18]. A particularly delicate issue in this context is that of separating the radiative component: finding that the charm PDF is nonzero at a low scale is not sufficient to argue that intrinsic charm has been identified.

Here we present a resolution of this four-decade-long conundrum by providing unambiguous evidence for intrinsic charm in the proton. This is achieved by means of a determination of the charm PDF (ref. [3]) from an extensive hard-scattering global dataset, using state-of-the-art perturbative QCD calculations[19], adapted to accommodate the possibility of massive quarks inside the proton[4,20,21], and sophisticated machine learning techniques[3,22,23]. This determination is performed at next-to-next-to-leading order (NNLO) in an expansion in powers of the strong coupling, $\alpha_s$, which represents the precision frontier for collider phenomenology.

The charm PDF determined in this manner includes a radiative component, and indeed it depends on the resolution scale: it is given in a four-flavour number scheme (4FNS), in which up, down, strange and charm quarks are subject to perturbative radiative corrections and mix with each other and the gluon as the resolution is increased. The intrinsic charm component can be disentangled from it as follows. First, we note that in the absence of an intrinsic component, the initial condition for the charm PDF is determined using perturbative matching conditions[24], computed up to NNLO in ref. [25], and recently (partly) extended up to next-to-next-to-next-to-leading order (N³LO; refs. [26–34]). These matching conditions determine the charm PDF in terms of the PDFs of the 3FNS, in which only the three lightest quark flavours are radiatively corrected. Hence, this perturbative charm PDF is entirely determined in terms of the three light quarks and antiquarks and the gluon. However, the 3FNS charm quark PDF needs not vanish: in fact, if the charm quark PDF in the 4FNS is freely parametrized and thus determined from the data[4], the matching conditions can be inverted. The 3FNS charm PDF thus obtained is then by definition the intrinsic charm PDF: indeed, in the absence of intrinsic charm it would vanish[21]. Thus, unlike the 4FNS charm PDF, which includes both an intrinsic and a radiative component, the 3FNS charm PDF is purely intrinsic. In this work we have performed this inversion at NNLO (ref. [25]) as well as at N³LO (refs. [26–34]), which—as we shall see—provides a handle on the perturbative uncertainty of the NNLO result.

Our starting point is the NNPDF4.0 global analysis[3], which provides a determination of the sum of the charm and anticharm PDFs,

namely $c^+(x, Q) \equiv c(x, Q) + \bar{c}(x, Q)$, in the 4FNS. This can be viewed as a probability density in $x$, the fraction of the proton momentum carried by charm, in the sense that the integral over all values of $0 \le x \le 1$ of $xc^+(x)$ is equal to the fraction of the proton momentum carried by charm quarks, although note that PDFs are generally not necessarily positive definite. Our result for the 4FNS $xc^+(x, Q)$ at the charm mass scale, $Q = m_c$ with $m_c = 1.51\,\text{GeV}$, is shown in Fig. 1 (left). The ensuing intrinsic charm is determined from it by transforming to the 3FNS using NNLO matching. This result is also shown in Fig. 1 (left). The bands indicate the 68% confidence level interval associated with the PDF uncertainties (PDFU) in each case. Henceforth, we will refer to the 3FNS $xc^+(x, Q)$ PDF as the intrinsic charm PDF.

The intrinsic (3FNS) charm PDF exhibits a characteristic valence-like structure at large $x$ peaking at $x \simeq 0.4$. Although intrinsic charm is found to be small in absolute terms (it contributes less than 1% to the proton total momentum), it is significantly different from zero. Note that the transformation to the 3FNS has little effect on the peak region, because there is almost no charm radiatively generated at such large values of $x$: in fact, a very similar valence-like peak is already found in the 4FNS calculation.

As at the charm mass scale the strong coupling $\alpha_s$ is rather large, the perturbative expansion converges slowly. To estimate the effect of missing higher-order uncertainties (MHOU), we have also performed the transformation from the 4FNS NNLO charm PDF determined from the data to the 3FNS (intrinsic) charm PDF at one order higher, namely at N³LO. The result is also shown Fig. 1 (left). Reassuringly, the intrinsic valence-like structure is unchanged. On the other hand, it is clear that for $x \lesssim 0.2$ perturbative uncertainties become very large. We can estimate the total uncertainty on our determination of intrinsic charm by adding in quadrature the PDFU and a MHOU estimated from the shift between the result found using NNLO and N³LO matching.

This procedure leads to our final result for intrinsic charm and its total uncertainty (shown in Fig. 1, right). The intrinsic charm PDF is found to be compatible with zero for $x \lesssim 0.2$: the negative trend seen in Fig. 1 with PDFU becomes compatible with zero only on inclusion of theoretical uncertainties. However, at larger $x$, even with theoretical uncertainties the intrinsic charm PDF differs from zero by about 2.5σ in the peak region. This result is stable on variations of dataset, methodology (in particular the PDF parametrization basis) and values of parameters (specifically the charm mass) of the standard model, as demonstrated in Supplementary Sections C and D.

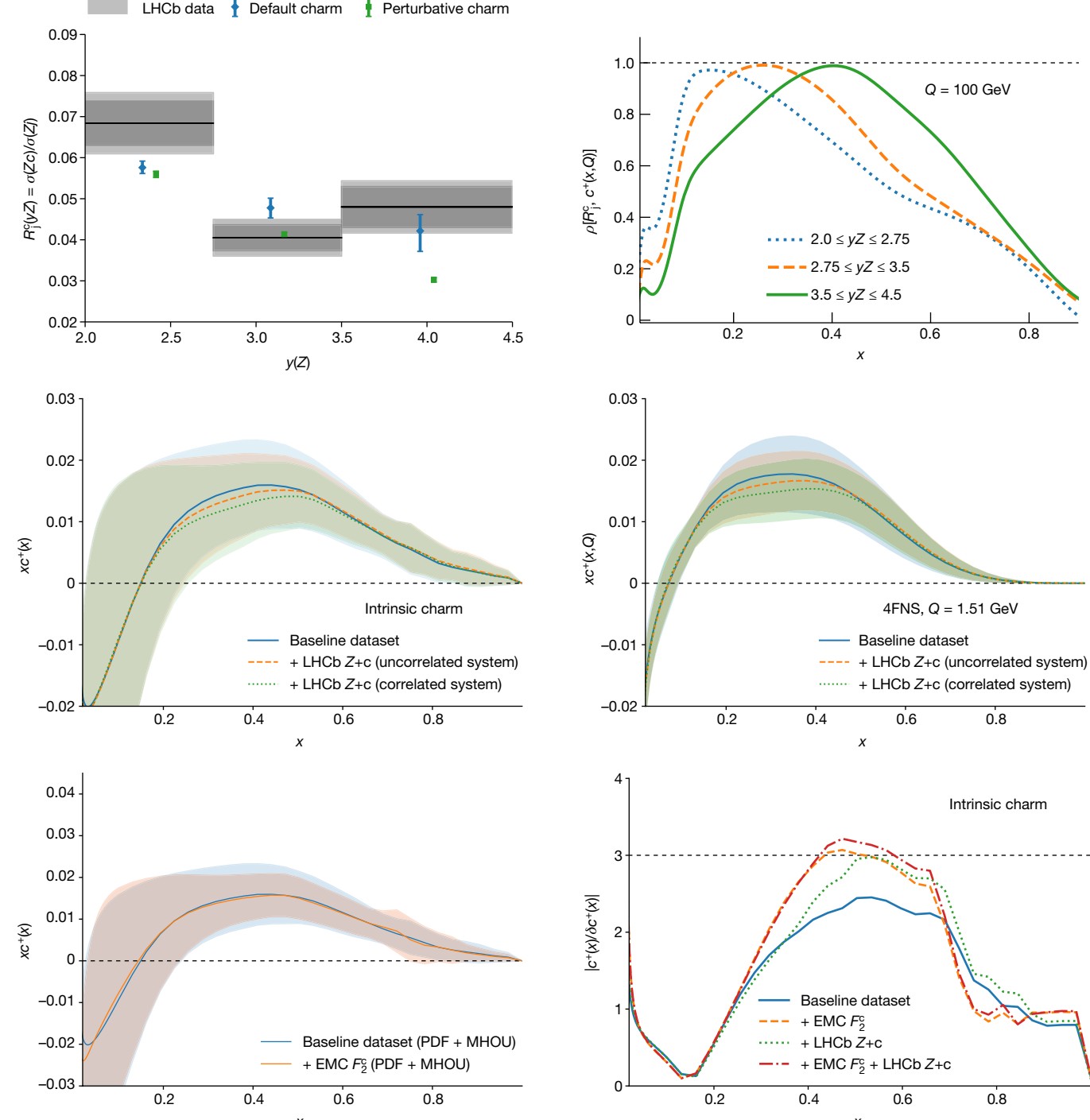

**Fig. 2 | Intrinsic charm and $Z$ + charm production at LHCb.** Top left, the LHCb measurements of $Z$-boson production in association with charm-tagged jets, $\mathcal{R}_j^c$, at $\sqrt{s}$ = 13 TeV, compared with our default prediction, which includes an intrinsic charm component, as well as with a variant in which we impose the vanishing of the intrinsic charm component. The thicker (thinner) bands in the LHCb data indicate the statistical (total) uncertainty, while the theory predictions include both PDFU and MHOU. Top right, the correlation coefficient between the charm PDF at $Q$ = 100 GeV in NNPDF4.0 and the LHCb measurements of $\mathcal{R}_j^c$ for the three $y_Z$ bins. The dotted horizontal line indicates

the maximum possible correlation. Centre, the charm PDF in the 4FNS (right) and the intrinsic (3FNS) charm PDF (left) before and after inclusion of the LHCb $Z$ + charm (c) data. Results are shown for both experimental correlation models discussed in the text. Bottom left, the intrinsic charm PDF before and after inclusion of the EMC charm structure function data. Bottom right, the statistical significance of the intrinsic charm PDF in our baseline analysis, compared to the results obtained also including the LHCb $Z$ + charm (with uncorrelated systematics) or the EMC structure function data, or both. The dotted horizontal line indicates the $3\sigma$ threshold.

Our determination of intrinsic charm can be compared to theoretical expectations. Subsequent to the original intrinsic charm model of ref. [1] (BHPS model), a variety of other models were proposed[5,35–38] (see ref. [2] for a review). Irrespective of their specific details, most models

predict a valence-like structure at large $x$ with a maximum located between $x \simeq 0.2$ and $x \simeq 0.5$, and a vanishing intrinsic component for $x \lesssim 0.1$. In Fig. 1 (right), we compare our result to the original BHPS model and to the more recent meson/baryon cloud model of ref. [5].

As these models predict only the shape of the intrinsic charm distribution, but not its overall normalization, we have normalized them by requiring that they reproduce the same charm momentum fraction as our determination. We find remarkable agreement between the shape of our determination and the model predictions. In particular, we reproduce the presence and location of the large-$x$ valence-like peak structure (with better agreement, of marginal statistical significance, with the meson/baryon cloud calculation), and the vanishing of intrinsic charm at small $x$. The fraction of the proton momentum carried by charm quarks that we obtain from our analysis, used in this comparison to models, is $(0.62 \pm 0.28)\%$ including PDFU alone (see Supplementary Section E for details). However, the uncertainty on inclusion of MHOU greatly increases, and we obtain $(0.62 \pm 0.61)\%$, due to the contribution from the small-$x$ region, $x \lesssim 0.2$, where the MHOU is very large (Fig. 1, right). Note that in most previous analyses[18] (see Supplementary Section F) intrinsic charm models (such as the BHPS model) are fitted to the data, with only the momentum fraction left as a free parameter.

We emphasize that in our analysis the charm PDF is entirely determined by the experimental data included in the PDF determination. The data with the most impact on charm are from recently measured LHC processes, which are both accurate and precise. As these measurements are made at high scales, the corresponding hard cross-sections can be reliably computed in QCD perturbation theory.

Independent evidence for intrinsic charm is provided by the very recent LHC beauty (LHCb) measurements of $Z$-boson production in association with charm-tagged jets in the forward region[6], which were not included in our baseline dataset. This process, when measured in terms of the ratio $\mathcal{R}_j^c$ of charm-tagged jets normalized to flavour-inclusive jets, is directly sensitive to the charm PDF (ref. [39]), and with LHCb kinematics also in the kinematic region where the intrinsic component is relevant. Following refs. [6,39], we have evaluated $\mathcal{R}_j^c$ at NLO (refs. [40,41]; see Supplementary Section G for details), both with our default PDFs that include intrinsic charm, and also with an independent PDF determination in which intrinsic charm is constrained to vanish identically, so charm is determined by perturbative matching (see Supplementary Section B).

In Fig. 2 (top left) we compare the LHCb measurements of $\mathcal{R}_j^c$, provided in three bins of the $Z$-boson rapidity $y_Z$, with the theoretical predictions based on both our default PDFs and the PDF set in which we impose the vanishing of intrinsic charm. In Fig. 2 (top right) we also show the correlation coefficient between the charm PDF at $Q = 100$ GeV and the observable $\mathcal{R}_j^c$, demonstrating how this observable is highly correlated to charm in a localized $x$ region that depends on the rapidity bin. It is clear that our prediction is in excellent agreement with the LHCb measurements, and in particular for the highest-rapidity bin, which is highly correlated to the charm PDF in the region of the observed valence peak $x \simeq 0.45$, the prediction obtained by imposing the vanishing of intrinsic charm undershoots the data at the $3\sigma$ level. Hence, this measurement provides independent direct evidence in support of our result.

We have also determined the impact of these LHCb $Z$ + charm measurements on the charm PDF. As the experimental covariance matrix is not available, we have considered two limiting scenarios in which the total systematic uncertainty is either completely uncorrelated ($\rho_{sys} = 0$) or fully correlated ($\rho_{sys} = 1$) between rapidity bins. The charm PDF in the 4FNS before and after inclusion of the LHCb data (with either correlation model), and the intrinsic charm PDF obtained from it, are shown in Fig. 2 (centre left and right, respectively). The bands account for both PDFU and MHOU. The results show full consistency: inclusion of the LHCb $\mathcal{R}_j^c$ data leaves the intrinsic charm PDF unchanged, while moderately reducing the uncertainty on it.

In the past, the main indication for intrinsic charm came from EMC data[42] on deep-inelastic scattering with charm in the final state[43]. These data are relatively imprecise, their accuracy has often been questioned, and they were taken at relatively low scales for which radiative corrections are large. For these reasons, we have not included them in our baseline analysis. However, it is interesting to assess the impact of their inclusion. Results are shown in Fig. 2 (bottom left), which shows the intrinsic charm PDF before and after inclusion of the EMC data. As in the case of the LHCb data, we find full consistency: unchanged shape and a moderate reduction of uncertainties.

We can summarize our results through their so-called local statistical significance (namely, the size of the intrinsic charm PDF in units of its total uncertainty). This is shown in Fig. 2 (bottom right) for our default determination of intrinsic charm, as well as after inclusion of either the LHCb $Z$ + charm or the EMC data, or both. We find a local significance for intrinsic charm at the $2.5\sigma$ level in the region $0.3 \lesssim x \lesssim 0.6$. This is increased to about $3\sigma$ by the inclusion of either the EMC or the LHCb data, and above if they are both included. The similarity of the impact of the EMC and LHCb measurements is especially remarkable in view of the fact that they involve very different physical processes and energies.

In summary, in this work we have presented evidence for intrinsic charm quarks in the proton. Our findings close a fundamental open question in the understanding of nucleon structure that has been hotly debated by particle and nuclear physicists for the past 40 years. By carefully disentangling the perturbative component, we obtain unambiguous evidence for intrinsic charm, which turns out to be in qualitative agreement with the expectations from model calculations. Our determination of the charm PDF, driven by indirect constraints from the latest high-precision LHC data, is perfectly consistent with direct constraints from both EMC charm production data taken 40 years ago, and very recent $Z$ + charm production data in the forward region from LHCb. Combining all data, we find a local significance for intrinsic charm in the large-$x$ region just above the $3\sigma$ level. Our results motivate further dedicated studies of intrinsic charm through a wide range of nuclear, particle and astroparticle physics experiments, such as those accessible at the High-Luminosity LHC (ref. [44]) and the fixed-target programmes of LHCb (ref. [45]) and A Large Ion Collider Experiment (ALICE)[46], to the Electron–Ion Collider, AFTER (ref. [47]), the Forward Physics Facility[48] and neutrino telescopes[49].

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

**The NNPDF Collaboration**

**Richard D. Ball[1], Alessandro Candido[2], Juan Cruz-Martinez[2], Stefano Forte[2], Tommaso Giani[3,4], Felix Hekhorn[2], Kirill Kudashkin[2], Giacomo Magni[3,4] & Juan Rojo[3,4]**

[1]The Higgs Centre for Theoretical Physics, University of Edinburgh, Edinburgh, UK. [2]Tif Lab, Dipartimento di Fisica, Università di Milano and INFN, Milan, Italy. [3]Department of Physics and Astronomy, Vrije Universiteit, Amsterdam, The Netherlands. [4]Nikhef Theory Group, Amsterdam, The Netherlands. ✉e-mail: j.rojo@vu.nl

## Methods

The strategy adopted in this work to determine the intrinsic charm content of the proton is based on the following observation. The assumption that there is no intrinsic charm amounts to the assumption that all 4FNS PDFs are determined[24] using perturbative matching conditions[25] in terms of 3FNS PDFs that do not include a charm PDF. However, these perturbative matching conditions are actually given by a square matrix that also includes a 3FNS charm PDF. Thus, the assumption of no intrinsic charm amounts to the assumption that if the 4FNS PDFs are transformed back to the 3FNS, the 3FNS charm PDF is found to vanish. Hence, intrinsic charm is by definition the deviation from zero of the 3FNS charm PDF (ref. [21]). Note that whereas the 3FNS charm PDF is purely intrinsic, the 4FNS charm PDF includes both an intrinsic and a perturbative radiative component, but the 4FNS intrinsic component is not equal to the 3FNS charm PDF, because matching conditions reshuffle all PDFs among each other.

Intrinsic charm can then be determined through the following two steps, summarized in Extended Data Fig. 1. First, all of the PDFs, including the charm PDF, are parametrized in the 4FNS at an input scale $Q_0$ and evolved using NNLO perturbative QCD to $Q \neq Q_0$. These evolved PDFs can be used to compute physical cross-sections, also at NNLO, which then are compared to a global dataset of experimental measurements. The result of this first step in our procedure is a Monte Carlo representation of the probability distribution for the 4FNS PDFs at the input parametrization scale $Q_0$.

Next, this 4FNS charm PDF is transformed to the 3FNS at some matching scale $Q_c$. Note that the choice of both $Q_0$ and $Q_c$ are immaterial—the former because perturbative evolution is invertible, so results for the PDFs do not depend on the choice of parametrization scale $Q_0$; the latter because the 3FNS charm is scale independent, so it does not depend on the value of $Q_c$. Both statements of course hold up to fixed perturbative accuracy, and are violated by missing higher order corrections. In practice, we parametrize PDFs at the scale $Q_0 = 1.65$ GeV and perform the inversion at a scale chosen equal to the charm mass $Q_c = m_c = 1.51$ GeV.

The scale-independent 3FNS charm PDF is then the sought-for intrinsic charm.

### Global QCD analysis

The 4FNS charm PDF and its associated uncertainties is determined by means of a global QCD analysis within the NNPDF4.0 framework. All PDFs, including the charm PDF, are parametrized at $Q_0 = 1.65$ GeV in a model-independent manner using a neural network, which is fitted to data using supervised machine learning techniques. The Monte Carlo replica method is deployed to ensure a faithful uncertainty estimate. Specifically, we express the 4FNS total charm PDF ($c^+ = c + \bar{c}$) in terms of the output neurons associated with the quark singlet $\Sigma$ and non-singlet $T_{15}$ distributions (see Section 3.1 of ref. [3]), as

$$xc^+(x, Q_0; \theta) = (x^{\alpha_\Sigma}(1-x)^{\beta_\Sigma}NN_\Sigma(x, \theta) - x^{\alpha_{T_{15}}}(1-x)^{\beta_{T_{15}}}NN_{T_{15}}(x, \theta))/4, \quad (1)$$

in which $NN_i(x, \theta)$ is the $i$th output neuron of a neural network with input $x$ and parameters $\theta$, and $(\alpha_i, \beta_i)$ are preprocessing exponents. A crucial feature of equation (1) is that no ad hoc specific model assumptions are used: the shape and size of $xc^+(x, Q_0)$ are entirely determined from experimental data. Hence, our determination of the 4FNS fitted charm PDF, and thus of the intrinsic charm, is unbiased.

The neural network parameters $\theta$ in equation (1) are determined by fitting an extensive global dataset that consists of 4,618 cross-sections from a wide range of different processes, measured over the years in a variety of fixed-target and collider experiments (see ref. [3] for a complete list). Extended Data Fig. 2 shows the kinematic coverage in the $(x, Q)$ plane covered by these cross-sections, in which $Q$ is the scale, and $x$ is the parton momentum fraction that corresponds to leading-order kinematics. Many of these processes provide direct or indirect sensitivity to the charm content of the proton. Particularly important constraints come from $W$ and $Z$ production from ATLAS, CMS and LHCb as well as from neutral and charged current deep-inelastic scattering structure functions from HERA. The 4FNS PDFs at the input scale $Q_0$ are related to experimental measurements at $Q \neq Q_0$ by means of NNLO QCD calculations, including the FONLL-C general-mass scheme for deep-inelastic scattering[20] generalized to allow for fitted charm[4].

We have verified (see Supplementary Section C and Section D) that the determination of 4FNS charm PDF equation (1) and the ensuing 3FNS intrinsic charm PDF are stable on variations of methodology (PDF parametrization basis), input dataset and values of parameters (the charm mass) of the standard model. We have also studied the stability of our results on replacing the current NNPDF4.0 methodology[3] with the previous NNPDF3.1 methodology[50]. The results are perfectly consistent. Indeed, the old methodology leads to larger uncertainties, corresponding to a moderate reduction of the local statistical significance for intrinsic charm, and to a central value that is within the smaller error band of our current result.

A determination in which the vanishing of intrinsic charm is imposed has also been performed. In this case, the fit quality markedly deteriorates: the values of the $\chi^2$ per data point of 1.162, 1.26 and 1.22 for total, Drell–Yan and neutral-current deep-inelastic scattering data, respectively, found when fitting charm, are increased to 1.198, 1.31 and 1.28 when the vanishing of intrinsic charm is imposed. The absolute worsening of the total $\chi^2$ when the vanishing of intrinsic charm is imposed is therefore of 166 units, corresponding to a $2\sigma$ effect in units of $\sigma_{\chi^2} = \sqrt{2n_{dat}}$.

### Calculation of the 3FNS charm PDF

The Monte Carlo representation of the probability distribution associated with the 4FNS charm PDF determined by the global analysis contains an intrinsic component mixed with a perturbatively generated contribution, with the latter becoming larger in the $x \lesssim 0.1$ region as the scale $Q$ is increased. To extract the intrinsic component, we transform PDFs to the 3FNS at the scale $Q_c = m_c = 1.51$ GeV using EKO, a Python open-source PDF evolution framework (see Supplementary Section A). In its current implementation, EKO performs QCD evolution of PDFs to any scale up to NNLO. For the sake of the current analysis, N³LO matching conditions have also been implemented, by using the results of refs. [26–34] for $\mathcal{O}(\alpha_s^3)$ operator matrix elements so that the direct and inverse transformations from the 3FNS to the 4FNS can be performed at one order higher. The N³LO contributions to the matching conditions are a subset of the full N³LO terms that would be required to perform a PDF determination to one higher perturbative order, and would also include N³LO contributions to QCD evolution that are unknown at present. Therefore, our results have NNLO accuracy, and we can use the N³LO contributions to the $\mathcal{O}(\alpha_s^3)$ corrections to the heavy quark matching conditions only as a way to estimate the size of the missing higher orders. Indeed, these corrections have a very significant impact on the perturbatively generated component (see Supplementary Section B). They become large for $x \lesssim 0.1$, which coincides with the region dominated by the perturbative component of the charm PDF, and are relatively small for the valence region where intrinsic charm dominates.

### *Z* production in association with charm-tagged jets

The production of $Z$ bosons in association with charm-tagged jets (or alternatively, with identified $D$ mesons) at the LHC is directly sensitive to the charm content of the proton through the dominant $gc \rightarrow Zc$ partonic scattering process. Measurements of this process at the forward rapidities covered by the LHCb acceptance provide access to the large-$x$ region where the intrinsic contribution is expected to dominate. This is in contrast with the corresponding measurements from ATLAS and CMS, which become sensitive to intrinsic charm only at rather larger values of $p_T^Z$ than those accessible experimentally at present.

We have obtained theoretical predictions for $Z$ + charm production at LHCb with NNPDF4.0, based on NLO QCD calculations using POWHEG-BOX interfaced to Pythia8 with the Monash 2013 tune for showering, hadronization and underlying event. Acceptance requirements and event selection follow the LHCb analysis, in which, in particular, charm jets are defined as those anti-$k_T R = 0.5$ jets containing a reconstructed charmed hadron. The ratio between charm-tagged and untagged $Z$ + jet events can then be compared with the LHCb measurements

$$\mathcal{R}_j^c(y_Z) \equiv \frac{N(\text{c−tagged jets}; y_Z)}{N(\text{jets}; y_Z)} = \frac{\sigma(pp \to Z + \text{charm jet}; y_Z)}{\sigma(pp \to Z + \text{jet}; y_Z)}, \quad (2)$$

as a function of the $Z$-boson rapidity $y_Z$ (see Supplementary Section G for details). The more forward the rapidity $y_Z$, the higher the values of the charm momentum $x$ being probed. Furthermore, we have also included the LHCb measurements in the global PDF determination by means of Bayesian reweighting (see Supplementary Section G).

## Data availability

Data are available from the corresponding author upon reasonable request. The experimental data used to perform the NNPDF4.0 analysis are available from https://docs.nnpdf.science/.

## Code availability

The analysis presented in this work has been carried out using two open-source software frameworks, NNPDF for the global PDF determination and EKO for the calculation of the 3FNS charm. These codes are publicly available from https://docs.nnpdf.science/ and https://eko.readthedocs.io/, respectively. Both the LHAPDF grids produced in this work and the version of EKO with the respective run cards used are available from http://nnpdf.mi.infn.it/nnpdf4-0-charm-study/.

50. Ball, R. D. et al. Parton distributions from high-precision collider data. *Eur. Phys. J. C* **77**, 663 (2017).

**Acknowledgements** We thank our colleagues of the NNPDF Collaboration for many illuminating discussions concerning the charm PDF. We are grateful to J. Blümlein for communicating Mathematica code with the results of refs. [26–34], to J. Ablinger for assistance in the implementation of the $\mathcal{O}(\alpha_s^3)$ calculation of the heavy-quark matching conditions, and to S. Zanoli for sharing her Mathematica implementation with us. We are grateful to R. Gauld for discussions, assistance and sharing his Pythia8 implementation for the calculation $Z$+charm production. We thank M. Guzzi and P. Nadolsky for discussions concerning intrinsic charm in the CT family of global PDF fits, and T. Hobbs and W. Melnitchouk for providing us with their predictions of the meson/baryon cloud model. We are grateful to T. Boettcher, P. Ilten and M. Williams for assistance with the LHCb $Z$+charm measurements. S.F., J.C.-M., F.H., A.C. and K.K. are supported by the European Research Council under the European Union's Horizon 2020 research and innovation programme (grant agreement number 740006). R.D.B. is supported by the UK Science and Technology Facility Council grant ST/P000630/1. J.R. and G.M. are partially supported by NWO (Dutch Research Council). T.G. is supported by NWO (Dutch Research Council) through an ENW-KLEIN-2 project.

**Author contributions** As is customary in high-energy physics, the authors are listed in alphabetical order. J.C.-M. is the main author of the new algorithm used in the NNPDF4.0 PDF determination. A.C., F.H. and G.M. developed the EKO code used to evaluate the 3FNS charm PDF, and specifically, G.M. implemented the matching conditions, with the help of K.K. for the implementation of some harmonic sums. T.G. performed the analysis of the LHCb $Z$+charm data. R.D.B. and S.F. designed the general procedure. J.R. coordinated the intrinsic charm determination and S.F. supervised the whole project. J.R. and S.F. wrote the paper and R.D.B. revised it. All authors discussed the results and their implications.

**Competing interests** The authors declare no competing interests.

**Additional information**
**Correspondence and requests for materials** should be addressed to Juan Rojo.

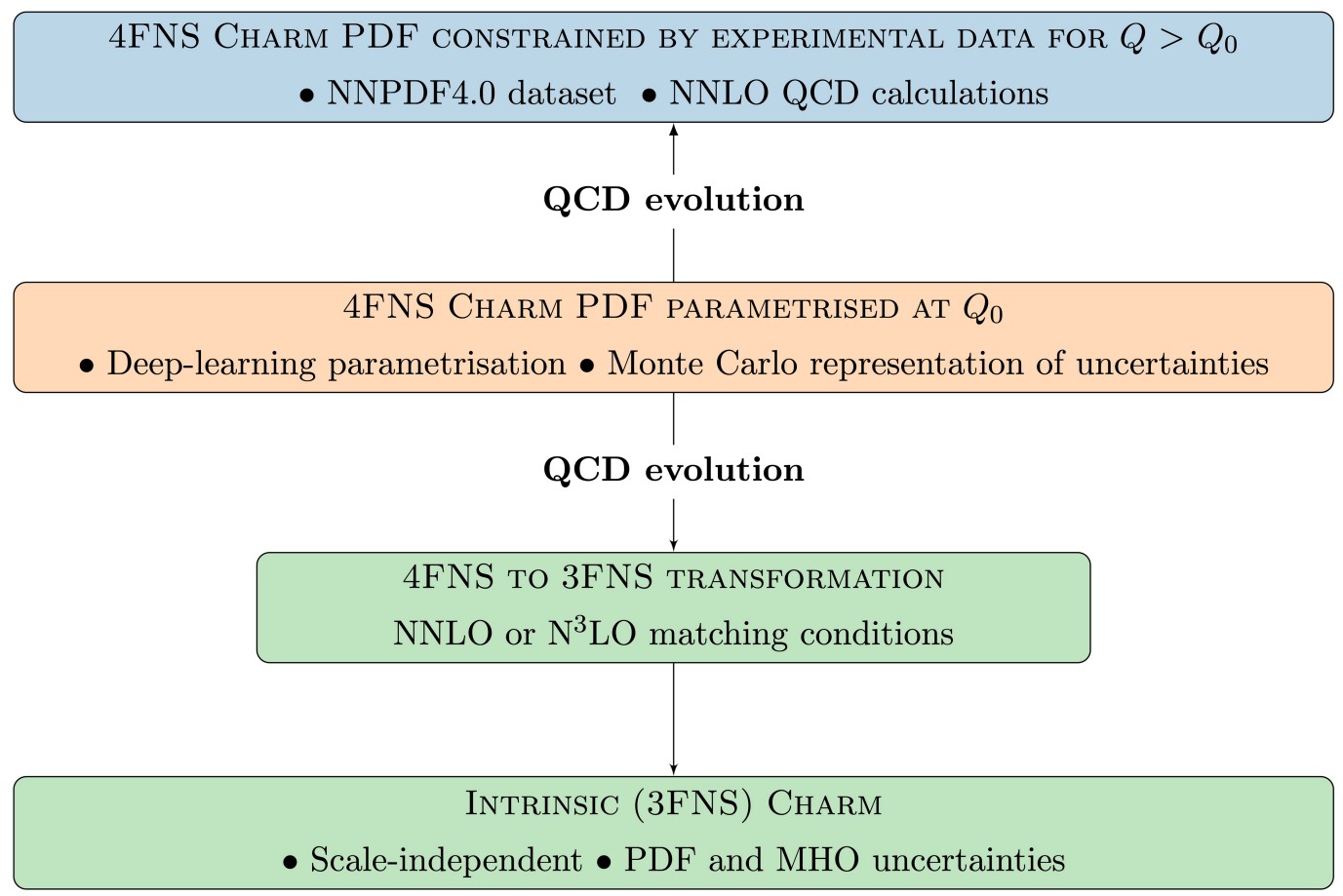

**Extended Data Fig. 1 | Evaluation of the charm PDF in the 3FNS.** The 4FNS charm PDF is parametrized at $Q_0$ and evolved to all $Q$, where it is constrained by the NNPDF4.0 global dataset. Subsequently, it is transformed to the 3FNS where (if nonzero) it provides the intrinsic charm component.

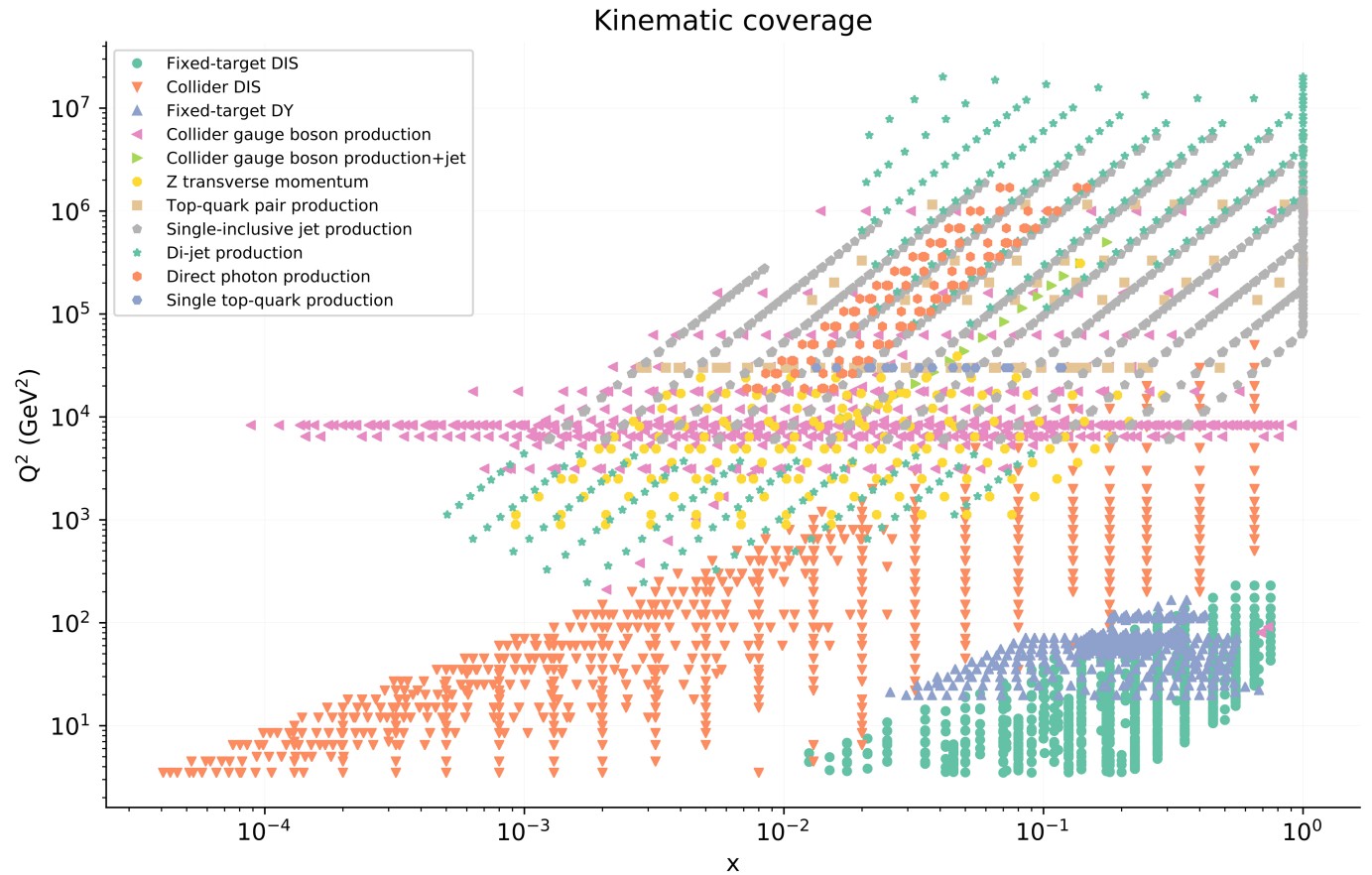

**Extended Data Fig. 2 | Kinematic coverage of the NNPDF4.0 determination.** The kinematic coverage in the $(x, Q)$ plane covered by the 4,618 cross-sections used for the determination of the charm PDF in the present work. These cross-sections have been classified into the main different types of processes entering the global analysis.