## [Peer Review File · Nature]

Manuscript Title: Evidence for intrinsic charm in the proton

Reviewer Comments & Author Rebuttals

Reviewer Reports on the Initial Version:

Referees' comments:

Referee #1 (Remarks to the Author):

This manuscript details a new evaluation of the charm distribution in the proton and provides evidence for a nonperturbative "intrinsic" charm component in the proton. The nearly 3 sigma effect seen is not enough to say that this is a "discovery" of intrinsic charm but it is good evidence and significant progress in a longstanding puzzle of QCD.

In the sense that intrinsic charm (IC in the following) was first predicted by Brodsky and collaborators around 1980, it is not very original. The topic of IC has been debated and various evidence has been presented over the years. Most of this evidence can be taken almost as anecdotal or has tried to be explained away by others, such as the high x and high Q² EMC data. However, in more recent years the subject has been taken up more seriously again and IC has been incorporated into a number of global analyses of the parton densities, including by some of these authors in the NNPDF collaboration, with varying, somewhat inconclusive results. What has changed in the last year is the Z+c jet data from the LHCb collaboration has showed solid evidence especially in their highest rapidity bin of the need for an IC contribution to explain their data. Indeed IC is undergoing a sort of renaissance right now. The subject matter is certainly worth publishing in Nature.

This work is very good. These authors have managed to separate out the perturbative charm contribution, which of course exists through radiative production in the proton, i.e. through gluon splitting. The analysis is thorough, rigorous and the result is robust. The authors have made a complete treatment of all uncertainties. The supplemental material contains useful information to back up the results presented in the main material. The manuscript is acceptable for publication.

I have only a few minor comments that should be considered prior to publication which I will now discuss.

There are several things in the first paragraph that I would like to mention. The second sentence says that the proton is a bound state of "two up and one down quark". It would seem more correct to say "two up quarks and one down quark" or "two up and one down quarks". That same sentence also refers to "an infinite number of quark-antiquark pairs" but doesn't mention there that these pairs are typically generated by an equally infinite number of gluons and it seems to me it should. I would also discourage the use of the possessive when it comes to a particle, e.g. find something to say other than "the proton's". Finally a bit later, I would somewhat object to the use of "static" to describe the nucleon wavefunction. The original IC papers do not mean to attribute IC to a nucleon at rest, they are still talking about the light cone and the momentum fraction x carried by IC is still

defined in the infinite momentum frame. The use of "static" seems to convey the wrong impression here.

The second paragraph under "Main" only refers to predictions of physics processes at the LHC. This is unnecessarily limiting and, indeed, the authors are more expansive in the final paragraph of "Main" where they also refer to future facilities like the EIC and neutrino telescopes, among others. It would make more sense to me to include some of these as motivation at the beginning. In addition, there are other, lower energy experiments proposed and in progress that could also shed light on IC, such as SMOG at LHCb and even lower fixed-target experiments that could probe IC at lower energies but high x . To only mention LHC at the beginning is a bit of a disservice and limits the potential interest at the beginning of the manuscript.

When the authors discuss the right-hand side of Fig. 1 and compare to BHPS and the meson-cloud results, they mention in passing that their result agrees slightly better with the meson-cloud model of IC. It would be worth offering some additional comments as to why they think this is so, i.e. some interpretation of why they reach this outcome that implies that the charm and anticharm distributions may be asymmetric.

In the main manuscript, the calculations in Fig. 1 are for $Q = mc = 1.51$ GeV but in the supplemental material the calculations in Secs. C and D are at $Q = 1.65$ GeV. This may be because that value of Q is just above the highest value of mc they study but they should actually say this if that's why. In addition, they only say that after they have shown results with this value of Q in two other figures in the section, before they even discuss the mass variation. It would be better if it was stated before they start showing all the other variations or move the discussion of the charm mass to first in that section. The comment in the caption of Fig. C.3. is rather cryptic and again refers to default but "default" is not crisply defined in the main section.

In Sec. B, the authors refer back to Fig. 2 (top left). They should say that this figure is in the main manuscript. (They do make that connection later but only at the end of Sec. G. They should do it the first time they refer to a figure in the main manuscript and make it clear that this is going to be the case in all instances.)

Also in Sec. B, in Fig. B.1. the authors show the power of the strong coupling constant in the legends of the figure but only mention NNLO and N3LO matching conditions. I would suggest using NNLO and N3LO in the legends instead, not only because that is what is used in the text but also because the power of α_s is hard to read in the legend of the figure.

The authors could better define NN_{c+} in Eq. (C.1).

In Fig. 2, the authors could be a bit clearer that their "default" result is the intrinsic charm because they don't really say much about the perturbative charm until the supplemental material.

In Fig. H.1. the authors are comparing their default charm to perturbative charm in the top four panels but in the last two the legend changes to "baseline charm". Is that actually different? The caption and the text seems to refer only to perturbative charm so I am not sure why the authors

suddenly change the legend in these two subfigures. I suggest they keep the legends consistent if they don't actually change.

All of these points are relatively minor and could easily be fixed to add to the otherwise very clear and cogent presentation that goes a long way toward a solution to this long-standing puzzle of QCD. The references are complete and the manuscript is generally easy to understand. The suggested corrections will only enhance its clarity. Once these suggestions have been incorporated, I can heartily recommend publication.

Referee #2 (Remarks to the Author):

This paper presents evidence of intrinsic charm of proton through global analysis in the framework of NNPDF4.0. This is a very important result to our understanding of proton. There are however a few points I would like to see addressed before recommending publication.

1. As shown in Figure 2 that the valence-like structure is there even before the inclusion of EMC F2C data and LHCb Z+c data. And as the authors mentioned that: "The data with the most impact on charm are from recently measured LHC processes, which are both accurate and precise." But I did not see the exact description on which specific data sets driven this evidence of intrinsic charm besides the EMC F2C and LHCb Z+c data in either the main text or the supporting material. Because this would be a milestone to our understanding of proton, it is important to know which data sets driven this conclusion.

2. As the authors mention that this result is stable upon variations of methodology. And the conclusion is the same either different charm mass, parametrisation on 4FNS charm at Q0, or even the inclusion of MHOU. But the authors did not mention the dependence on methodology of global analysis. I was wondering do we get similar 3 sigma evidence by using the framework of NNPDF3.1 but with data sets of NNPDF4.0?

3. The intrinsic charm shown in the Figure 1 shows a very clear valence-like structure for $x c^+$ at $x \sim 0.4$. And it also shows relatively large $x c^+$ for $x \rightarrow 1$. It seems to me that this intrinsic charm is compatible with strange PDF for $x \rightarrow 1$. This is somewhat different from our expectation; how do we understand the behavior of intrinsic charm for $x \rightarrow 1$ from this result?

Similarly, as the authors mention: "This can be viewed as a probability density in x , the fraction of the proton's momentum carried by charm, in the sense that the integral over all values of $0 < x < 1$ of $x c^+$ is equal to the fraction of the proton momentum carried by charm quarks." While in figure 1, we also see obvious negative behavior of $x c^+$ for $x < 0.2$. How do we understand the behavior of intrinsic charm with negative value in small x region?

4. As the authors mention that the momentum fraction of intrinsic charm is $(0.62 \pm 0.28)\%$ including PDF uncertainties only, and it is $(0.62 \pm 0.61)\%$ including MHOU. In the other words, when looking at the momentum fraction, the evidence of intrinsic charm become 2 sigma or less. It would be great to see the authors address it a bit more when we consider it as evidence of intrinsic charm.

Referee #3 (Remarks to the Author):

A. Summary of the key results

The manuscript presents a novel extraction of intrinsic charm quarks in the proton from a large collection of experimental data, finding evidence for intrinsic charm at the 3-sigma level. The extraction method takes advantage of machine learning techniques, specifically neural networks. The particularly novel aspect of the extraction method is that it disentangles intrinsic charm from radiatively generated charm by cleverly inverting a typical matching procedure between charm parton distribution function determinations allowing 3 (up, down, strange) vs. 4 (up, down, strange, charm) flavors. The shape and magnitude of the intrinsic charm distribution thus extracted are compared to and found to agree with models, further supporting the validity of the extraction. The specific impact of two experimental data sets, an older F_2^c measurement from EMC and a recent Z+charm-jet measurement from LHCb, is additionally examined. Consistent results for the charm PDF are extracted with and without these data sets; their simultaneous inclusion increases the significance of the evidence for intrinsic charm in the proton to the 3-sigma level.

B. Originality and significance

The work is original; it brings a novel method and the largest-ever experimental data set to a decades-old question that has remained of interest, reaching the strongest conclusion on the question thus far. While 5-sigma is often informally considered the threshold for a discovery claim and only 3-sigma significance is reported, the fact that the magnitude and shape of the extracted intrinsic charm distribution agree with model predictions provides additional evidence to support the discovery claim.

C. Data & methodology: validity of approach, quality of data, quality of presentation

While I am not directly involved in performing PDF extractions myself and thus not familiar with all subtleties, the NNPDF Collaboration approach to standard PDF extraction is well established and highly regarded. The novel approach invoked here to disentangle intrinsic from perturbative charm is clever and makes sense. The dominant sources of uncertainty on the extracted distribution are considered and discussed. However, the presentation does not currently address one point that I feel is key to the message of the paper and will not be familiar to many readers of Nature. All extracted charm PDFs are shown in the plots to go negative below some x value, while the reader is told to view them as probability densities. The model calculations of the intrinsic charm PDF on the right-hand-side of Fig. 1, in contrast, show the more intuitive behavior of being positive semidefinite and going to zero as x goes to zero. The negative region in the extracted fits will impact the total integral over x ; this integral is part of the main message of the paper. The meaning of the negative region of the extracted PDFs should be addressed explicitly in the text. Also related to this is the statement on p. 3 that the intrinsic charm PDF “is compatible with zero for $x < \sim 0.2$.” The left side of Fig. 1 shows three curves going strongly negative, and the uncertainty bands shown do not include zero. Again related, on p. 4 “the vanishing of intrinsic charm at small- x ” is mentioned.

In a similar vein in Fig. 1, even though the total and intrinsic charm curves at large x are consistent within uncertainties, given that the intrinsic charm must be a subset of the total when integrated over x , it would be valuable to address the fact that the total curve is below the intrinsic curves at large x .

D. Appropriate use of statistics and treatment of uncertainties

As noted above, the statistical methods of the NNPDF Collaboration are well established, and the treatment of uncertainties here is generally appropriate. One point that could be clarified is why the N3LO match is not preferred over the NNLO match.

E. Conclusions: robustness, validity, reliability

The robustness of the extraction is checked for NNLO vs. N3LO matching and found to be reliable for the large- x region of interest. As mentioned above, the interpretation or significance of the regions where the extractions go negative should be addressed explicitly in the text. I understand the results to be valid.

F. Suggested improvements: experiments, data for possible revision

I only have suggestions to improve the clarity of the text and its accessibility to a wide audience, included below.

G. References: appropriate credit to previous work?

I did not note any key references that were omitted but am not deeply familiar with the extensive literature on intrinsic charm from the past 40 years. On p. 2, I note that in the sentence “it is a well-known fact [8, 10]...” that both references cited are by authors of the present manuscript. I agree with the claim being made but feel it would be appropriate to include also one or more references independent of the current authors. (I also note that the recent LHCb results in Ref. 6 are now published in PRL.)

H. Clarity and context: lucidity of abstract/summary, appropriateness of abstract, introduction and conclusions

The text of the manuscript could be changed in several places to improve accessibility to a wide range of readers. In the introduction on the top of p. 2 where the proton’s wave-function is mentioned, a very brief discussion of why we do not know the proton’s wave-function, or at least an explicit statement that it is not known, would be helpful. I have found many physicists working outside of hadronic physics to take it for granted that we must know the proton’s wave-function.

p. 2 – “is barely larger than that of the proton.” – m_c is >50% more than the mass of the proton, so it is a very large relative difference, even though ~ 0.5 GeV when considered in the context of LHC energies is small. I think the relevant context for the reader would be that the charm quark is the only heavy quark light enough for people to consider whether it has an intrinsic component in the proton. The b mass immediately jumps up several more GeV.

p. 2 – “diametrically opposite claims, from excluding momentum fractions carried by intrinsic charm larger than 0.5% at the 4-sigma level to allowing up to a 2% charm momentum fraction.” – These are not “diametrically opposite claims”—one allows intrinsic charm up to 0.5%, the other up to 2%.

p. 2 – $\alpha_s(Q)$ – Q is not defined.

Fig. 1 right – The light blue uncertainty band is nearly invisible on my printout. It would also be better to differentiate the curve styles on the left-hand side.

The language surrounding Fig. 1 (legend, caption, and body of text) is not consistent, leading to potential confusion. In particular, it should be made consistently clear that the “4FNS charm” is “total” charm.

p. 4 – “Independent evidence for intrinsic charm is provided by the very recent LHCb measurements” – It should be stated explicitly that this LHCb data was not included in the default fit.

In Fig. 2, it was striking to me to see that the 1983 EMC measurement has more or less the same impact as the very recent LHCb measurement. The authors can consider commenting on this.

Fig. 4 (Methods section) – Are there so many jet and dijet data points with $x > 0.9$? This looks strange to me, but perhaps it is correct.

p. 11 (Methods section) – “the fit quality significantly deteriorates” – The χ^2 values per data point, while they increase when the vanishing of intrinsic charm is imposed, do not seem to me to be at all unreasonable for a global PDF fit. It is only a ~3% increase. Some evidence for why this change is significant should be given, as it impacts the conclusion of the paper.

Author Rebuttals to Initial Comments:

We would like to thank all the referees for their appreciation and for their very useful and constructive criticisms. We address below each of the points raised by the referees in turn, and describe the actions that have been taken.

Together with the revised submission of our manuscript, we have also uploaded in the system a pdf file in which additions and removals of text to the original manuscript have been respectively highlighted in blue and red.

Referee #1

This manuscript details a new evaluation of the charm distribution in the proton and provides evidence for a nonperturbative "intrinsic" charm component in the proton. The nearly 3 sigma effect seen is not enough to say that this is a "discovery" of intrinsic charm but it is good evidence and significant progress in a longstanding puzzle of QCD.

In the sense that intrinsic charm (IC in the following) was first predicted by Brodsky and collaborators around 1980, it is not very original. The topic of IC has been debated and various evidence has been presented over the years. Most of this evidence can be taken almost as anecdotal or has tried to be explained away by others, such as the high x and high Q^2 EMC data. However, in more recent years the subject has been taken up more seriously again and IC has been incorporated into a number of global analyses of the parton densities, including by some of these authors in the NNPDF collaboration, with varying, somewhat inconclusive results. What has changed in the last year is the $Z+c$ jet data from the LHCb collaboration has showed solid evidence especially in their highest rapidity bin of the need for an IC contribution to explain their data. Indeed IC is undergoing a sort of renaissance right now. The subject matter is certainly worth publishing in Nature.

This work is very good. These authors have managed to separate out the perturbative charm contribution, which of course exists through radiative production in the proton, i.e. through gluon splitting. The analysis is thorough, rigorous and the result is robust. The authors have made a complete treatment of all uncertainties. The supplemental material contains useful information to back up the results presented in the main material. The manuscript is acceptable for publication.

I have only a few minor comments that should be considered prior to publication which I will now discuss.

- 1. There are several things in the first paragraph that I would like to mention. The second sentence says that the proton is a bound state of "two up and one down quark". It would seem more correct to say "two up quarks and one down quark" or "two up and one down quarks". That same sentence also refers to "an infinite number of quark-antiquark pairs" but doesn't mention there that these pairs are typically generated by an equally infinite number of gluons and it seems to me it should. I would also discourage the use of the possessive when it comes to a particle, e.g. find something to say other than "the proton's". Finally a bit later, I would somewhat object to the use of "static" to describe the nucleon wavefunction. The original IC papers do not mean to attribute IC to a nucleon at rest, they are still talking about the light cone and the momentum fraction x carried by IC is still defined in the infinite momentum frame. The use of "static" seems to convey the wrong impression here.*

We thank the referee for suggesting these improvements.

Action: we have changed the wording of the first paragraph and implemented all the corrections suggested by the referee.

- 2. The second paragraph under "Main" only refers to predictions of physics processes at*

the LHC. This is unnecessarily limiting and, indeed, the authors are more expansive in the final paragraph of "Main" where they also refer to future facilities like the EIC and neutrino telescopes, among others. It would make more sense to me to include some of these as motivation at the beginning. In addition, there are other, lower energy experiments proposed and in progress that could also shed light on IC, such as SMOG at LHCb and even lower fixed-target experiments that could probe IC at lower energies but high x . To only mention LHC at the beginning is a bit of a disservice and limits the potential interest at the beginning of the manuscript.

We agree with both points raised by the referee: that our results are relevant for many other facilities and experiments, not just the LHC, and that several lower energy experiments are relevant for intrinsic charm determination.

Action: we have changed the sentence at the beginning of the second paragraph of the Main Text and we have added further references to future experiments in the last sentence of the Main Text.

3. When the authors discuss the right-hand side of Fig. 1 and compare to BHPS and the meson-cloud results, they mention in passing that their result agrees slightly better with the meson-cloud model of IC. It would be worth offering some additional comments as to why they think this is so, i.e. some interpretation of why they reach this outcome that implies that the charm and anticharm distributions may be asymmetric.

Discriminating between different non-perturbative models of proton structure is beyond the scope of our work: in fact, it is unclear whether the somewhat better agreement of our results of the meson cloud model (MCM) in comparison to the BHPS model is significant. Specifically, whereas it is true that the intrinsic charm asymmetry vanishes in the BHPS model while it is nonzero in the MCM, it is unclear whether this plays any role in leading to this better agreement, given that in our analysis we do not allow for a charm-anticharm asymmetry. In order to investigate this question, it would be interesting to repeat our analysis while also allowing for a nonzero charm-anticharm asymmetry. Answering this interesting question is left for future work.

Action: we have modified the reference to the quality of the agreement with the MCM in order to emphasize that it is unclear whether the difference is significant.

4. In the main manuscript, the calculations in Fig. 1 are for $Q = m_c = 1.51$ GeV but in the supplemental material the calculations in Secs. C and D are at $Q = 1.65$ GeV. This may be because that value of Q is just above the highest value of m_c they study but they should actually say this if that's why. In addition, they only say that after they have shown results with this value of Q in two other figures in the section, before they even discuss the mass variation. It would be better if it was stated before they start showing all the other variations or move the discussion of the charm mass to first in that section. The comment in the caption of Fig. C.3. is rather cryptic and again refers to default but "default" is not crisply defined in the main section.

Indeed, as correctly pointed out by the referee, the reason why in the Supporting Information some plots are shown at $Q = 1.65$ GeV is because showing 4FNS charm PDF plots at $Q = 1.65$ GeV is necessary in order to display the results for the complete range of m_c variations considered in our work, in particular since the highest value is $m_c = 1.64$ GeV. To clarify this point, as suggested by the referee in the revised version of the paper we mention at the beginning of Sect. C that in that section the charm PDF is displayed at $Q = 1.65$ GeV so that results for fit variants with different m_c value can be displayed at a common scale.

We also note that in the case of Sect. D, all results shown are in the 3FNS and hence scale independent.

Action: We have added to the beginning of Sect. C the justification of the choice of scale $Q = 1.65$ GeV for the plots of the 4FNS charm PDF, namely so that results for fit variants with different m_c values can be displayed at a common scale. We have at the beginning of Sect. B an explicit statement that “default” will always refer to the result shown in Fig. 1 (right).

5. In Sec. B, the authors refer back to Fig. 2 (top left). They should say that this figure is in the main manuscript. (They do make that connection later but only at the end of Sec. G. They should do it the first time they refer to a figure in the main manuscript and make it clear that this is going to be the case in all instances.)

Action: we have implemented the suggestion from the referee to add the explicit reference to the main manuscript the first time we refer to Fig. 2 (top left).

6. Also in Sec. B, in Fig. B.1. the authors show the power of the strong coupling constant in the legends of the figure but only mention NNLO and N3LO matching conditions. I would suggest using NNLO and N3LO in the legends instead, not only because that is what is used in the text but also because the power of α_s is hard to read in the legend of the figure.

Action: the legends in Fig. B.1 have been updated to read “NNLO match” and “N³LO match” as suggested by the referee.

7. The authors could better define NN_{c+} in Eq. (C.1).

Action: we have added a statement linking Eq. (C.1) with Eq. (1), indicating that $NN_{c+}(x)$ represents the value of the output neuron associated to the parametrization of the total charm PDF.

8. In Fig. 2, the authors could be a bit clearer that their “default” result is the intrinsic charm because they don’t really say much about the perturbative charm until the supplemental material.

Action: we have modified both the caption to Fig. 2 and the text referring to it, now explicitly stating that by default we mean our results which include an intrinsic charm component.

9. In Fig. H.1. the authors are comparing their default charm to perturbative charm in the top four panels but in the last two the legend changes to “baseline charm”. Is that actually different? The caption and the text seems to refer only to perturbative charm so I am not sure why the authors suddenly change the legend in these two sub-figures. I suggest they keep the legends consistent if they don’t actually change.

We thank the referee for spotting this typo.

Action: the orange bands are now correctly labeled as “perturbative charm” in the bottom panels of Fig. H.1.

All of these points are relatively minor and could easily be fixed to add to the otherwise very clear and cogent presentation that goes a long way toward a solution to this long-standing puzzle of QCD. The references are complete and the manuscript is generally easy to understand. The suggested corrections will only enhance its clarity. Once these suggestions have been incorporated, I can heartily recommend publication.

Referee #2

This paper present evidence of intrinsic charm of proton through global analysis in the framework of NNPDF4.0. This is an very important result to our understanding of proton. There are however a few points I would like to see addressed before recommending publication.

1. *As shown in Figure 2 that the valence-like structure is there even before the inclusion of EMC F_2^c data and LHCb $Z+c$ data. And as the authors mentioned that: “The data with the most impact on charm are from recently measured LHC processes, which are both accurate and precise.” But I did not see the exact description on which specific data sets driven this evidence of intrinsic charm besides the EMC F_2^c and LHCb $Z+c$ data in either the main text or the supporting material. Because this would be a milestone to our understanding of proton, it is important to know which data sets driven this conclusion.*

We thank the referee for asking this interesting question. In order to answer it, we have produced four extra plots in which the intrinsic charm PDF, as determined from the baseline dataset, is compared to determinations in which some data are excluded, to be added to those already shown in Fig. D.1 of the Supporting Information. These plots are obtained from four new fits: a fit without any W, Z production data from ATLAS and CMS, a fit without jet data, a fit without $Z p_T$ measurements, and a fit without HERA structure function data.

These new plots are shown in Fig. 1 below, and have now been added to Fig. D.1 of the manuscript. These plots reinforce the conclusion that hadron collider data are more important than deep-inelastic data, that among these LHCb data are playing a dominant role, and show that among the remaining hadron collider data jet data also play a non-negligible role.

Action: The plots of Fig. 1 of this document have been added to Fig. D.1 of the paper, together with the corresponding discussion.

2. *As the authors mention that this result is stable upon variations of methodology. And the conclusion is the same either different charm mass, parametrisation on 4FNS charm at Q_0 , or even the inclusion of MHO. But the authors did not mention the dependence on methodology of global analysis. I was wondering do we get similar 3 sigma evidence by using the framework of NNPDF3.1 but with data sets of NNPDF4.0?*

In order to address this point, we have repeated our determination of intrinsic charm, with exactly the same dataset, but now using the NNPDF3.1 fitting methodology. The 4FNS charm PDF at $Q = 1.65$ GeV found in this way is compared to our default result in Fig. 2. We find that results obtained with either methodology are perfectly consistent, but the previous methodology is less precise, leading to a reduction of the local statistical significance for intrinsic charm from the 2.5σ level of our baseline result to about 1.5σ .

Action: we have added a brief discussion of this issue to the methods section (last two sentences of next-to-last paragraph of “Global QCD analysis” in Methods).

3. *The intrinsic charm shown in the Figure 1 shows a very clear valence-like structure for xc^+ at $x \sim 0.4$. And it also shows relatively large xc^+ for $x \rightarrow 1$. It seems to me that this intrinsic charm is compatible with strange PDF for $x \rightarrow 1$. This is somewhat different from our expectation; how do we understand the behavior of intrinsic charm for $x \rightarrow 1$ from this result?*

The referee’s statement is correct. This is demonstrated in Fig. 3 below, where the total charm and strangeness of our baseline analysis in the 3FNS are compared at $Q = 1.5$ GeV (note that while 3FNS charm is scale independent, the strangeness PDF depends on

Figure 1. Comparison of the baseline determination of the intrinsic (3FNS) charm PDF with fit variants based on data subsets: without HERA structure functions, without ATLAS W, Z inclusive data, without jet production measurements, and without $Z p_T$ data.

Q). It is clear that at large values of x , $x \gtrsim 0.4$, the total strange and charm PDFs are quite similar. This agreement of large x strange and charm is surprising to the extent that intrinsic charm is surprising: and indeed, our result is in agreement with intrinsic charm models. So our conclusion here is that this observation suggests that at large- x PDFs are dominated by intrinsic light-cone dynamics.

4. Similarly, as the authors mention: “This can be viewed as a probability density in x , the fraction of the proton’s momentum carried by charm, in the sense that the integral over all values of $0 < x < 1$ of xc^+ is equal to the fraction of the proton momentum carried by charm quarks.” While in figure 1, we also see obvious negative behavior of xc^+ for $x < 0.2$. How do we understand the behavior of intrinsic charm with negative value in small x region?

These are very important points. We agree with the referee that the statement that the PDF can be viewed as a probability density without qualification can be misleading, especially to non-experts, since PDFs beyond leading order can be negative. As for the reason why charm goes negative at small x , some relevant observations are that even purely perturbative charm in the 4FNS gives negative at small x , and furthermore, that if $N^3\text{LO}$ matching conditions are used a very different (and even more negative) small- x behavior is found. This suggests that the negative small x behavior of the charm PDF might be an artifact of the truncation of an unstable perturbative expansion of both matching conditions and evolution equations at small x , that would be cured by small- x

Figure 2. The 4FNS charm PDF at $Q = 1.65$ GeV, comparing the baseline results of the present work (based on the NNPDF4.0 methodology) with their counterparts from a determination based on the NNPDF3.1 fitting methodology and the same dataset as in the NNPDF4.0 analysis.

Figure 3. Comparison of the 3FNS total charm and total strangeness PDFs in our baseline analysis.

resummation. Indeed, once we account for an estimate of the uncertainty involved in this truncation, by comparing NNLO and N³LO matching, we find that within uncertainties small- x charm is very compatible with zero, as one might expect and as predicted by models.

Action: we have qualified the sentence on probability densities in order to make clear that PDFs are not necessarily positive. We have added an explicit comment to the statement

that IC is compatible with zero at small x (last paragraph of Pag. 3) stating that it is the theoretical uncertainty that makes the negative behavior compatible with zero.

5. *As the authors mention that the momentum fraction of intrinsic charm is $(0.62 \pm 0.28)\%$ including PDF uncertainties only, and it is $(0.62 \pm 0.61)\%$ including MHOUs. In the other words, when looking at the momentum fraction, the evidence of intrinsic charm become 2 sigma or less. It would be great to see the authors address it a bit more when we consider it as evidence of intrinsic charm.*

While the momentum fraction carried by charm quarks has been traditionally adopted as one of the main estimators to quantify to which extent IC is allowed/excluded by the experimental data, one of the main findings of our analysis is that actually once MHOUs are accounted for the information that this quantity provides on IC is limited. The reason is the large perturbative corrections that affect the transformation from the 4FNS scheme (where the charm PDF has both intrinsic and perturbative components) to the 3FNS one (where the perturbative component has been removed), as quantified by the shift between the NNLO and N3LO transformations. These MHOUs are large for $x < 0.2$, and hence while they do not affect the local evidence of IC, they lower the “integrated” evidence since the momentum integral requires knowledge of the charm PDF all the way down to $x = 0$. This is why the momentum fraction is not the most suitable estimator to assess the presence/absence of intrinsic charm, and why local measures such as that displayed in Fig. 2 (bottom right) in our manuscript are more appropriate.

The role that the large MHOUs affecting the 3FNS charm PDF for $x < 0.2$ play in determining the corresponding momentum fraction can be highlighted by evaluating the truncated charm momentum fraction defined as

$$[c](x_{\min}) = \int_{x_{\min}}^1 dx xc^+(x, Q^2). \quad (1)$$

which for the 3FNS intrinsic charm is scale independent. We show in Fig. 4 below this truncated integral as a function of the lower integration limit x_{\min} for our baseline determination. From this result we see that provided one restricts the momentum integral to $x_{\min} \sim 0.2$, the statistical significance is similar as that from the local pull, a significance which is then washed out by the large MHOUs present in the region $x \lesssim 0.2$.

Action: We have added to the SI Sect. E the plot with the truncated momentum integral evaluated as a function of the lower integration limit x_{\min} together with the corresponding discussion (and of this section). We have also added an explicit statement that while the total momentum fraction is traditionally adopted as a measure of intrinsic charm in our analysis we find that the information that it provides is limited unless one looks at the partial fraction.

Figure 4. The value of the truncated charm momentum integral, Eq. (1) in this document, as a function of the lower integration limit x_{\min} for our baseline determination of the 3FNS intrinsic charm PDF. We display separately the PDF and the total (PDF+MHOU) uncertainties.

Referee #3

A. Summary of the key results.

The manuscript presents a novel extraction of intrinsic charm quarks in the proton from a large collection of experimental data, finding evidence for intrinsic charm at the 3-sigma level. The extraction method takes advantage of machine learning techniques, specifically neural networks. The particularly novel aspect of the extraction method is that it disentangles intrinsic charm from radiatively generated charm by cleverly inverting a typical matching procedure between charm parton distribution function determinations allowing 3 (up, down, strange) vs. 4 (up, down, strange, charm) flavors. The shape and magnitude of the intrinsic charm distribution thus extracted are compared to and found to agree with models, further supporting the validity of the extraction. The specific impact of two experimental data sets, an older F_2^C measurement from EMC and a recent Z+charm-jet measurement from LHCb, is additionally examined. Consistent results for the charm PDF are extracted with and without these data sets; their simultaneous inclusion increases the significance of the evidence for intrinsic charm in the proton to the 3-sigma level.

B. Originality and significance

The work is original; it brings a novel method and the largest-ever experimental data set to a decades-old question that has remained of interest, reaching the strongest conclusion on the question thus far. While 5-sigma is often informally considered the threshold for a discovery claim and only 3-sigma significance is reported, the fact that the magnitude and shape of the extracted intrinsic charm distribution agree with model predictions provides additional evidence to support the discovery claim.

C. Data & methodology: validity of approach, quality of data, quality of presentation

While I am not directly involved in performing PDF extractions myself and thus not familiar with all subtleties, the NNPDF Collaboration approach to standard PDF extraction is well established and highly regarded. The novel approach invoked here to disentangle intrinsic from perturbative charm is clever and makes sense. The dominant sources of uncertainty on the extracted distribution are considered and discussed. However, the presentation does not currently address one point that I feel is key to the message of the paper and will not be familiar to many readers of Nature. All extracted charm PDFs are shown in the plots to go negative below some x value, while the reader is told to view them as probability densities. The model calculations of the intrinsic charm PDF on the right-hand-side of Fig. 1, in contrast, show the more intuitive behavior of being positive semidefinite and going to zero as x goes to zero. The negative region in the extracted fits will impact the total integral over x ; this integral is part of the main message of the paper. The meaning of the negative region of the extracted PDFs should be addressed explicitly in the text. Also related to this is the statement on p. 3 that the intrinsic charm PDF “is compatible with zero for $x \lesssim 0.2$.” The left side of Fig. 1 shows three curves going strongly negative, and the uncertainty bands shown do not include zero. Again related, on p. 4 “the vanishing of intrinsic charm at small- x ” is mentioned.

We agree with the referee that these points deserve clarification: indeed, the same points were also raised by Referee 2, points 4-5. First, the statement that PDFs “can be viewed” as probability densities could be misleading to many readers of Nature and needs qualification - PDFs, unlike probability densities, can be negative. Second, the negative behavior of the charm PDF at small x becomes compatible with zero once uncertainties related to missing higher order corrections are accounted for. This means that this negative behavior could well be a byproduct of the perturbative truncation and could go away upon resummation of small x perturbative corrections. Third, this uncertainty in the small x region is large enough to render the full momentum integral useless as a measure of intrinsic charm: even though it seems unlikely and unappealing, a large negative contribution from the small x region cannot be excluded, so the

evidence for intrinsic charm only comes from the local significance in the large x region.

Actions: see the actions indicated in our reply to points 4 and 5 of Referee 2.

In a similar vein in Fig. 1, even though the total and intrinsic charm curves at large x are consistent within uncertainties, given that the intrinsic charm must be a subset of the total when integrated over x , it would be valuable to address the fact that the total curve is below the intrinsic curves at large x .

The referee raises a subtle point here: the 4FNS charm PDF includes both a perturbative and an intrinsic component, while in the 3FNS perturbative charm vanishes and the charm PDF is purely intrinsic. However, the 3FNS charm PDF cannot be identified with the 4FNS intrinsic component, because 3FNS and 4FNS are related by matching conditions that reshuffle all PDFs with each other. Rather, the 4FNS intrinsic component is the difference between the total, and the 4FNS perturbative component. The 4FNS perturbative component, in turn, is what is found when applying perturbative matching to a vanishing 3FNS intrinsic charm, as discussed in Sect. B of the Supplementary Information. This 4FNS perturbative component is shown in Fig. B.1, and, as the referee correctly expected, it is well below the total.

While we agree with the referee that this issue should be addressed, we feel that it is somewhat too technical for the main text, and we have accordingly added a discussion in the Supplementary Information, Section B.

Action: We have added a discussion at the end of the first paragraph in the Methods section. Also, we have added a sentence (penultimate sentence of the last paragraph of Section B in the Supplementary Information) in which we discuss the decomposition of the 4FNS result into a perturbative and intrinsic component.

D. Appropriate use of statistics and treatment of uncertainties

As noted above, the statistical methods of the NNPDF Collaboration are well established, and the treatment of uncertainties here is generally appropriate. One point that could be clarified is why the N3LO match is not preferred over the NNLO match.

The reason is that the matching conditions are just a subset of the full N3LO corrections: specifically, the N3LO contributions to evolution equations are unknown. Therefore by including a subset of N3LO terms we can get an estimate of the size of missing higher orders, but we cannot obtain full N3LO accuracy, and thus our central result is consistently obtained at NNLO.

Action: We have added a discussion to the Methods section (“Calculation of the 3FNS charm PDF”) in which we explain that N3LO matching terms are just a subset of the full N3LO contributions.

E. Conclusions: robustness, validity, reliability

The robustness of the extraction is checked for NNLO vs. N3LO matching and found to be reliable for the large- x region of interest. As mentioned above, the interpretation or significance of the regions where the extractions go negative should be addressed explicitly in the text. I understand the results to be valid.

We have addressed the issue of negative charm, as discussed in point C above.

F. Suggested improvements: experiments, data for possible revision

I only have suggestions to improve the clarity of the text and its accessibility to a wide audience, included below.

G. References: appropriate credit to previous work?

I did not note any key references that were omitted but am not deeply familiar with the extensive literature on intrinsic charm from the past 40 years. On p. 2, I note that in the sentence “it is a well-known fact [8, 10]. . .” that both references cited are by authors of the present manuscript. I agree with the claim being made but feel it would be appropriate to include also one or more references independent of the current authors. (I also note that the recent LHCb results in Ref. 6 are now published in PRL.)

Action: we have added the reference to two more review papers by other authors, and updated all references to published versions.

H. Clarity and context: lucidity of abstract/summary, appropriateness of abstract, introduction and conclusions.

The text of the manuscript could be changed in several places to improve accessibility to a wide range of readers. In the introduction on the top of p. 2 where the proton’s wave-function is mentioned, a very brief discussion of why we do not know the proton’s wave-function, or at least an explicit statement that it is not known, would be helpful. I have found many physicists working outside of hadronic physics to take it for granted that we must know the proton’s wave-function.

Action: we have added an explicit statement to the fourth sentence of the paper (before “Main”), and we have modified the sentence referred to by the referee.

1. *p. 2 – “is barely larger than that of the proton.” – m_c is >50% more than the mass of the proton, so it is a very large relative difference, even though 0.5 GeV when considered in the context of LHC energies is small. I think the relevant context for the reader would be that the charm quark is the only heavy quark light enough for people to consider whether it has an intrinsic component in the proton. The b mass immediately jumps up several more GeV.*

Action: we have changed “barely larger” to “the same order of magnitude”.

2. *p. 2 – “diametrically opposite claims, from excluding momentum fractions carried by intrinsic charm larger than 0.5% at the 4-sigma level to allowing up to a 2% charm momentum fraction.” – These are not “diametrically opposite claims”—one allows intrinsic charm up to 0.5%, the other up to 2%.*

Action: we have replaced “diametrically opposite claims” with “disparate claims”.

3. *p. 2 – $\alpha_s(Q)$ – Q is not defined.*

The referee is absolutely right: this is clearly not the right place to discuss the scale dependence of the coupling constant.

Action: we have replaced $\alpha_s(Q)$ with α_s .

4. *Fig. 1 right – The light blue uncertainty band is nearly invisible on my printout. It would also be better to differentiate the curve styles on the left-hand side.*

Action: In Fig. 1 (left) we use now different line styles (solid, dashed, dotted) for each of the three bands shown. In Fig. 1 (right), following the suggestion from the referee, we marked the error bands with darker colors

5. *The language surrounding Fig. 1 (legend, caption, and body of text) is not consistent, leading to potential confusion. In particular, it should be made consistently clear that the “4FNS charm” is “total” charm.*

As discussed above (point C), calling the 4FNS “total charm” is potentially misleading as it could lead the reader to think that this is the sum of the 3FNS intrinsic charm, plus a radiation (perturbative) component. We realize that in this respect having called c^+ the “total” charm PDF might have been misleading because by “total” we meant “quark plus antiquark”, while it might have been construed as “perturbative plus intrinsic”. However, we agree with the referee that the nomenclature Fig. 1 is not fully consistent with the text and potentially confusing.

Action: we have changed the caption to Fig. 1 and the text of the paragraph that first refers to Fig. 1 in order to make clear that what we call “intrinsic charm” and the 3FNS are one and the same, and that the 4FNS corresponds to a different result that includes both an intrinsic and a radiation component.

6. *p. 4 – “Independent evidence for intrinsic charm is provided by the very recent LHCb measurements” – It should be stated explicitly that this LHCb data was not included in the default fit.*

Action: we have added a clause at the end of this sentence, explicitly stating that the LHCb Z +charm data is not included in the baseline dataset.

7. *In Fig. 2, it was striking to me to see that the 1983 EMC measurement has more or less the same impact as the very recent LHCb measurement. The authors can consider commenting on this.*

We fully agree with the referee here, and we had tried to convey this with the sentence at the end of the penultimate paragraph before the Acknowledgments.

Action: we have strengthened the sentence at the end of the penultimate paragraph before the Acknowledgments, now emphasizing that the agreement between data from different processes at different energies is especially remarkable.

8. *Fig. 4 (Methods section) – Are there so many jet and dijet data points with $x \lesssim 0.9$? This looks strange to me, but perhaps it is correct.*

The reason for this is that while there is a single value of Q^2 that contributes to each data point, there is no single value of x : hence, in this plot, which is for illustrative purposes, we show the value or values of x that gives the dominant contribution. For processes like W production, at lowest perturbative order, there is only one value of x for each data point. So in that case we plot this value, even though in the NNLO computation that is not the only value that contributes. For jet production already at leading order a range of values of x is allowed, so in that case we plot the extremes of the allowed leading-order range.

9. *p. 11 (Methods section) – “the fit quality significantly deteriorates” – The χ^2 values per data point, while they increase when the vanishing of intrinsic charm is imposed, do not seem to me to at all unreasonable for a global PDF fit. It is only a 3% increase. Some evidence for why this change is significant should be given, as it impacts the conclusion of the paper.*

It is true that the deterioration of the χ^2 per data point is small in percentage, however this correspond to a statistically significant difference in total χ^2 . With $n_{\text{dat}} = 4618$ data, the deterioration from $\chi^2/n_{\text{dat}} = 1.162$ to $\chi^2/n_{\text{dat}} = 1.198$ corresponds to an increase of the χ^2 value by $\Delta\chi^2 = 166$. Because the standard deviation of the χ^2 distribution is $\sqrt{2n_{\text{dat}}} = 96$, this is about a 2σ effect, which is indeed similar to the size of the evidence we have for intrinsic charm.

Action: we have added to the “Methods” section the value of the absolute change in χ^2 when the vanishing of intrinsic charm is imposed as well as the number of sigmas that this variation corresponds to.

We hope that after having addressed the points raised by the referees, our revised manuscript can be considered as suitable for publication in Nature.

Reviewer Reports on the First Revision:

Referees' comments:

Referee #1 (Remarks to the Author):

Having reviewed the manuscript previously, I will only address any new issues I have noticed in the revised manuscript, which I find suitable for publication in Nature. The authors have addressed my concerns and, in my opinion, also those of the other two referees. My comments here are all relatively minor, except for the last one.

On page 3 and several other places throughout the manuscript, the authors mention the charm PDF having an intrinsic and a radiation part. I believe a better word would be "radiative" instead of "radiation".

The abbreviations MHO and PDFU appear in the caption and legend of Fig. 1 respectively. While MHO is only defined in the text below the figure, PDFU (presumably PDF uncertainty) is not explicitly defined. The abbreviations should be defined where they first appear in the text.

On page 4, the authors refer to models being "fitted to the data". I believe "fit to the data" is more widely used but this is not so important.

On page 12, the authors changed "corresponds" to "correspond" but the first was correct.

At the end of the first full paragraph on page 13, "a 2 sigma effects" should be "a 2 sigma effect".

In the first paragraph of part B in the supplemental material, "in the sequel" should be something rather like "in the following". Sequel is not really used correctly here.

My most important comment in the main manuscript is that Ref. [45] is inappropriate for the usage where it appears. That paper is about antiproton production in the LHCb fixed-target program. There is a publication about charm at fixed target by the collaboration. That publication is the one that should be referenced.

Once these rather small things have been addressed, the manuscript should be suitable for publication.

Referee #2 (Remarks to the Author):

The authors present an very important work on showing the evidence of intrinsic charm from the PDFs global analysis side. With the new updates which addressed all the issues I asked, I am happy to suggest the publication of this work.

An additional minor suggestion of mine is that, it maybe great to include the Fig.2 (in the replying

mail), which shows the intrinsic charm by using default NNPDF4.0 and NNPDF3.1 with 4.0 data sets, in the Supplementary Information. I believe it will make the discussion even complete.

Referee #3 (Remarks to the Author):

The authors have adequately addressed all of my comments on the previous version, and I recommend publication after one additional point is addressed related to Fig. D1 and its discussion in the new supplemental material.

Fig. D1 – This is a valuable addition to the paper but leaves me unclear on some points. The panel comparing the baseline dataset to the one with no LHCb W, Z data seems to indicate that the LHCb W, Z data suggest a smaller intrinsic charm contribution than the rest of the baseline data would imply. I also note that the uncertainties as plotted don't seem to be significantly reduced by adding the LHCb W, Z data. Similarly, the extraction using the DIS-only dataset in the top right panel indicates that the pp data in general suggest a smaller intrinsic charm contribution than DIS measurements do; however, with a bit less impact, the bottom left panel comparing the baseline dataset to that with no jets instead shows that jet data (from pp only or also from HERA?) imply a slightly larger intrinsic charm contribution than the rest of the data. Currently the updated text says, "Comparison of all the dataset variations shows that, in terms of their impact on intrinsic charm, hadron collider data are generally more important than deep-inelastic scattering data, that among the former the LHCb inclusive W, Z data are playing a dominant role, and that jet observables also play a non-negligible role." From these plots showing the dataset variations, in particular looking at the curves and uncertainty bands for DIS-only, collider-only, and baseline without LHCb W,Z, I don't see how the hadron collider data are significantly more important than the DIS data. The DIS data indicate a larger intrinsic charm contribution than the baseline and lead to an uncertainty that's only modestly larger for the $x > 0.2$ region of interest. The collider-only data suggest a smaller intrinsic charm contribution that's in fact consistent with zero within the (larger) uncertainties. Or perhaps "collider dataset" includes also HERA? This should be clarified. I do agree that within the collider data, the LHCb inclusive W and Z measurements have a large impact; however, their impact appears to reduce the significance of the claim for intrinsic charm from >4 sigma without those data to $< \sim 3$ sigma once they are included. Overall Fig. D.1 leaves me with the impression that a stronger claim for intrinsic charm, farther from zero and with similar uncertainties to the baseline results presented here, could have been made prior to the turn-on of the LHC, and I am not sure how the reader is supposed to interpret that. Potential higher-twist effects in the DIS observables are currently mentioned, possibly implying additional sources of uncertainty on the DIS data, but no potential uncertainties on the DIS data beyond what is plotted are explicitly discussed. In general I feel that the discussion of the dependence on the input dataset should be more thorough.

Author Rebuttals to First Revision:

We would like to thank the three referees for their appreciation and for recommending the publication of our revised paper in Nature. We address here the final remarks raised by the referees.

Referee #1

Having reviewed the manuscript previously, I will only address any new issues I have noticed in the revised manuscript, which I find suitable for publication in Nature. The authors have addressed my concerns and, in my opinion, also those of the other two referees. My comments here are all relatively minor, except for the last one.

On page 3 and several other places throughout the manuscript, the authors mention the charm PDF having an intrinsic and a radiation part. I believe a better word would be "radiative" instead of "radiation".

Action: we agree with the referee and have changed "radiation" by "radiative" when referring to the charm component.

The abbreviations MHOU and PDFU appear in the caption and legend of Fig. 1 respectively. While MHOU is only defined in the text below the figure, PDFU (presumably PDF uncertainty) is not explicitly defined. The abbreviations should be defined where they first appear in the text.

We have defined the abbreviation "PDF uncertainties" (PDFU) the first time it appears in the text.

On page 4, the authors refer to models being "fitted to the data". I believe "fit to the data" is more widely used but this is not so important.

No action taken here.

On page 12, the authors changed "corresponds" to "correspond" but the first was correct.

Action: fixed to original verbal tense.

At the end of the first full paragraph on page 13, "a 2 sigma effects" should be "a 2 sigma effect".

Action: typo corrected.

In the first paragraph of part B in the supplemental material, "in the sequel" should be something rather like "in the following". Sequel is not really used correctly here.

Action: followed the suggestion from the referee.

My most important comment in the main manuscript is that Ref. [45] is inappropriate for the usage where it appears. That paper is about antiproton production in the LHCb fixed-target program. There is a publication about charm at fixed target by the collaboration. That publication is the one that should be referenced.

Action: we have replaced ref [45] with the appropriate reference describing the measurement of charm production in the fixed target mode by the LHC collaboration, namely "First Measurement of Charm Production in its Fixed-Target Configuration at the LHC", arXiv:1810.07907, Phys. Rev. Lett. (122) 132002 (2019).

Once these rather small things have been addressed, the manuscript should be suitable for publication.

Referee #2

The authors present an very important work on showing the evidence of intrinsic charm from the PDFs global analysis side. With the new updates which addressed all the issues I asked, I am happy to suggest the publication of this work.

An additional minor suggestion of mine is that, it maybe great to include the Fig. 2 (in the replying mail), which shows the intrinsic charm by using default NNPDF4.0 and NNPDF3.1 with 4.0 data sets, in the Supplementary Information. I believe it will make the discussion even complete.

Action: following the suggestion from the referee, we have added Fig. 2 from our previous "referee-reply.pdf" file to the Supplementary Information of the paper, specifically to Sect C "Stability of the 4FNS charm PDF", together with the corresponding discussion.

Referee #3

The authors have adequately addressed all of my comments on the previous version, and I recommend publication after one additional point is addressed related to Fig. D1 and its discussion in the new supplemental material.

Fig. D1 – This is a valuable addition to the paper but leaves me unclear on some points. The panel comparing the baseline dataset to the one with no LHCb W, Z data seems to indicate that the LHCb W, Z data suggest a smaller intrinsic charm contribution than the rest of the baseline data would imply. I also note that the uncertainties as plotted don't seem to be significantly reduced by adding the LHCb W, Z data. Similarly, the extraction using the DIS-only dataset in the top right panel indicates that the pp data in general suggest a smaller intrinsic charm contribution than DIS measurements do; however, with a bit less impact, the bottom left panel comparing the baseline dataset to that with no jets instead shows that jet data (from pp only

or also from HERA?) imply a slightly larger intrinsic charm contribution than the rest of the data. Currently the updated text says, “Comparison of all the dataset variations shows that, in terms of their impact on intrinsic charm, hadron collider data are generally more important than deep-inelastic scattering data, that among the former the LHCb inclusive W , Z data are playing a dominant role, and that jet observables also play a non-negligible role.” From these plots showing the dataset variations, in particular looking at the curves and uncertainty bands for DIS-only, collider-only, and baseline without LHCb W, Z , I don’t see how the hadron collider data are significantly more important than the DIS data. The DIS data indicate a larger intrinsic charm contribution than the baseline and lead to an uncertainty that’s only modestly larger for the $x > 0.2$ region of interest. The collider-only data suggest a smaller intrinsic charm contribution that’s in fact consistent with zero within the (larger) uncertainties. Or perhaps “collider dataset” includes also HERA? This should be clarified. I do agree that within the collider data, the LHCb inclusive W and Z measurements have a large impact; however, their impact appears to reduce the significance of the claim for intrinsic charm from ≥ 4 sigma without those data to $\lesssim 3$ sigma once they are included. Overall Fig. D.1 leaves me with the impression that a stronger claim for intrinsic charm, farther from zero and with similar uncertainties to the baseline results presented here, could have been made prior to the turn-on of the LHC, and I am not sure how the reader is supposed to interpret that. Potential higher-twist effects in the DIS observables are currently mentioned, possibly implying additional sources of uncertainty on the DIS data, but no potential uncertainties on the DIS data beyond what is plotted are explicitly discussed. In general I feel that the discussion of the dependence on the input dataset should be more thorough.

We would like to thank the referee for raising this point, because it made us realize that the layout of of Fig. D1 in the previous revision of the paper could lead to some confusion. This is due to the fact that the eight plots shown in that figure did not all have the same scale on the y axis. In the current revision of the Supplementary Information, we have replaced Fig. D1 with Fig. 1 in this document, which is identical in terms of content, but now all plots have exactly the same range on the y axis. We believe that the new version of the figures clarifies the situation and addresses several of the questions raised by the referee, in that it is now much easier to gauge the relative impact in intrinsic charm of the various input datasets that we consider.

This said, coming to the various points raised by the referee:

- We confirm that the “collider dataset” includes the HERA DIS structure functions, since HERA is a lepton-proton collider. This is standard notation used in the literature, and we have added a note to the discussion to make this definition unambiguous. Whenever we refer to non-HERA collider data, we use the terminology “hadron collider data”.
- We note that while the DIS-only intrinsic charm has a larger central value, the uncertainties are also larger, by up to a factor 2 as compared to the baseline result. In addition, the DIS-only fit is less stable, in that it shows larger point-by-point fluctuations, to the reduced experimental constraints. Therefore, even though in a very narrow range of x the local significance of intrinsic charm from the DIS-only fit is comparable to that of the global fit, if only these data were available it would be difficult to exclude that the intrinsic charm signal was just a local fluctuation. Finally, comparison with the collider-only fit, which includes HERA data, shows that it is the DIS data at large- x from old fixed target experiments that have an impact on intrinsic charm. These, as also noted by the referee, come from a kinematic region characterized by large missing higher order corrections and higher twist effects, thereby making a determination from DIS data less reliable.
- Concerning the role of hadron collider data, it is clear that some of them (LHCb W and Z production) lead to a stronger signal and others (jets) to a slightly weaker one, as it is of course to be expected for a global dataset. However, all hadron collider data lead to a

consistent and stable signal, to be contrasted to the significantly less stable signal seen in the DIS-only fit.

Action: We have updated Fig. D1 in the Supplementary Information with a version (identical content) that uses the same range for the y axis in all panels and hence facilitates the comparison of the impact of the different datasets. We have explicitly stated that the definition of the “collider-only” dataset includes the HERA structure function data. We have expanded the sentence discussing the role of different data, stressing the somewhat unstable nature of the DIS-only fit, and the fact that fixed-target DIS data are affected by potential sources of theory uncertainties, such as higher twists.

Figure 1. The revised version of Fig. D1 in the Supplementary Information.